# Calibration of Compartmental Epidemiological Models via Graybox Bayesian Optimization

Puhua Niu
*Electrical & Computer Engineering*
*Texas A&M University*
College Station, TX, USA
niupuhua.123@tamu.edu

Byung-Jun Yoon
*Electrical & Computer Engineering*
*Texas A&M University*
College Station, TX, USA
*Computational Science Initiative*
*Brookhaven National Laboratory*
Upton, NY, USA
bjyoon@ece.tamu.edu

Xiaoning Qian
*Electrical & Computer Engineering*
*Texas A&M University*
College Station, TX, USA
*Computer Science & Engineering*
*Texas A&M University*
College Station, TX, USA
*Computational Science Initiative*
*Brookhaven National Laboratory*
Upton, NY, USA
xqian@ece.tamu.edu

*Abstract*—In this study, we focus on developing efficient calibration methods via Bayesian decision-making for the family of compartmental epidemiological models. The existing calibration methods usually assume the compartmental model is *cheap* in terms of its output and gradient evaluation, which may not hold in practice when extending them to more general settings. Therefore, we introduce model calibration methods based on a "graybox" Bayesian optimization (BO) scheme, more efficient calibration for general epidemiological models. This approach uses Gaussian processes as a surrogate to the expensive model, and leverages the functional structure of the compartmental model to enhance calibration performance. Additionally, we develop model calibration methods via a decoupled decision-making strategy for BO, which further exploits the decomposable nature of the functional structure. The calibration efficiencies of the multiple proposed schemes are evaluated based on various data generated by a compartmental model mimicking real-world epidemic processes. Experimental results demonstrate that our proposed graybox variants of BO schemes can further improve the calibration performance measured by the logarithm of mean square errors and achieve faster performance convergence in terms of BO iterations. We anticipate that the proposed calibration methods can be extended to enable fast calibration of more complex epidemiological models, such as the agent-based models.

*Index Terms*—Compartmental Model, Bayesian Optimization, Model Calibration, Gaussian Process, Knowledge Gradient

## I. Introduction

Pandemics like the recent coronavirus disease (COVID-19) have shown enormous impacts on public health worldwide. The development of computational epidemiological models is crucial for gaining better quantitative understanding of the disease spread and enabling swift decision-making to design effective mitigation strategies. In fact, well-calibrated epidemiological models can serve as a useful guide for forecasting and quantifying the epidemic risk and implementing effective public health measures to control the spread of epidemic diseases [2], [8].

This work was supported by the U.S. Department of Energy (DOE) Office of Science under Award KJ0403010/FWP CC132.

Compartmental models, such as the the commonly adopted SIR (Susceptible-Infectious-Recovered) model, constitute an important family of population-based epidemiological models. Models in this family take the form of Ordinary Differential Equation (ODE) systems that capture the general trends and dynamics in the subpopulations represented by different compartments. In this work, we focus on this family of models while considering a new "Quarantined" state represented by an additional compartment. This new state aims at better mimicking the dynamic trajectories of the epidemic spreads, typical in airborne diseases such as seasonal flu and COVID-19, resulting in a four-compartment SIQR model [23].

To accurately model real-world physical processes, it is crucial for computer models to fine-tune their model parameters based on observed data to capture inherent attributes of the physical processes, which can not be directly or easily measured by means of physical experiments. For instance, material properties, such as porosity and permeability are important computer inputs in computational material simulations, which cannot be measured directly in physical experiments. In the applied mathematics and computational science literature, the methods used to identify those parameters are called model calibration techniques and the resulting parameters are called calibration parameters [20], [25]. The basic idea of calibration is to find the combination of the calibration parameters, under which the simulated computer outputs align with the observed physical data. A well-calibrated SIQR model may allow people to make more accurate predictions of pandemic risk and enable them to make better-informed decisions on how to mitigate.

The existing method often assumes that the computer model is *cheap* [30]. This means that: the explicit form of the optimization objective $f(x)$ for calibration parameters $x$ is known and we can evaluate the first- or second-order derivatives of $f(x)$ with respect to (w.r.t.) $x$. For an ODE system, deriving the explicit form of $f(x)$ and $\nabla_x f(x)$ can be infeasible when the ODE is non-linear. Recent advances show that sensitivity analysis allows us to evaluate the gradient $\nabla_x f(x)$

and perform the gradient-descent optimization method for the estimation of $x$ [12], [18]. Correspondingly, physics-informed machine learning [19] is drawing increasing attention as it integrates data and mathematical models into deep neural networks or other regression models by enforcing fundamental physical laws.

In real-world scenarios when calibrating a computer simulation software, we are usually blind to the explicit forms of both $f(x)$ and $\nabla_x f(x)$ and we may be able to access only the output of $f(x)$. We classify this type of computer models as *expensive*. Hence, it is valuable to pay attention to the calibration problem for such *expensive* computer models. Bayesian Optimization (BO) is one conventional approach for such scenario when evaluating $f(x)$ is expensive, where we use a surrogate model to approximate $f(x)$ and the optimization process is sequential. In each BO round, we make a *Bayesian decision* so that some expected utility is maximized and then the corresponding output is queried from the computer model based on this decision to refine the surrogate model to guide the optimization of the objective function $f(x)$ [14], [17].

This paper concentrates specifically on the calibration of the SIQR model based on BO and is organized as follows. First, we explain the configurations of the SIQR model and the model calibration setup. Then we introduce our calibration methods based on a "graybox BO" approach, instead of the usual "blackbox BO", where the experts' prior knowledge about the computer models is integrated into the BO formulation to improve its optimization performance. Finally, we explain how the graybox BO works for decoupled decision-making, accompanied by a new acquisition function. In our experiments, we evaluate the performance of our proposed graybox BO-based model calibration methods and compare them with traditional blackbox BO-based implementations to demonstrate the benefits of integrating prior model knowledge into the calibration process.

## II. PRELIMINARIES

### A. SIQR model

In this model, people in a given population are classified into four compartments: *Susceptible (S)*, *Infectious (I)*, *Recovered (R)*, and *Quarantined (Q)*. Those who have not been infected are considered susceptible. Transmission occurs between susceptible and infected individuals. The number of symptomatic patients, equivalent to the number of infected individuals, decreases through treatment and/or quarantine. Recovered individuals are assumed to remain immune to further infection. Therefore, all people will be in the recovered group after an adequate period. The dynamics of the SIQR model are given by the following set of ordinary differential equations (ODEs) [23]:

$$\begin{cases} \nabla_t S(t) = & -\beta(t)S(t) \\ \nabla_t I(t) = & \beta(t)S(t) - \lambda(t)I(t) - \gamma I(t) \\ \nabla_t R(t) = & \gamma I(t) + \delta Q(t) \\ \nabla_t Q(t) = & \lambda(t)I(t) - \delta Q(t) \end{cases} \quad (1)$$

where $S(t)$, $I(t)$, $Q(t)$ and $R(t) \in [0, N]$ are the populations of compartments 'S', 'I', 'Q' and 'R' respectively. It can be verified that the ODEs guarantee the conservation of the population, $S(t)+I(t)+Q(t)+R(t) = N$. A schematic illustration of such an SIQR model is shown in Figure 1. We note that there are many compartmental epidemiological models based on different extensions of the original SIR model [10], [24]. While we focus on SIQR model calibration in this paper, the presented model calibration methods can be applied to all these variants in a straightforward manner. More importantly, calibration of more complicated epidemiology models based on agent-based models (ABMs) can also leverage our proposed methods if their critical parameters can be identified, where calibration efficiency is even more important due to their significantly higher computational complexity compared to the ODE-based compartmental models [1], [4], [9].

### B. Model Calibration

Most of the existing calibration methods [11], [13], [26], [28], [29], [31] follow or adapt from the Bayesian approach pioneered by Kennedy and O'Hagan (KOH) [20]. In practice, we can only collect epidemiological data in the occurrence of an epidemic. In other words, we can observe the population's dynamic trajectory *only once*, making model calibration more challenging than in other statistical inference or machine learning tasks. Supposing one such trajectory with time length $T$ are observed, we denote the observed epidemiological data as $\{\mathbf{d}^t = (d_1^t, ..., d_4^t)\}_{t=1}^T$ and make the following assumption:

$$\mathbf{d}^t = \zeta^t + \epsilon \approx \eta^t(x) + \epsilon, \quad (2)$$

where $\zeta$ is the unknown ground-truth underlying epidemic process, $\eta$ is a computer model (e.g., SIQR model) that we would like to calibrate to simulate the behavior of $\zeta$ and fit the observed epidemiological data, $x \in \mathbb{R}^D$ is the $D$-dimensional *calibration* parameter vector of $\eta$, and $\epsilon \sim \mathcal{N}(0, \text{Diag}(\mathbf{1}))$ represents the observation noise. In this work, $\eta^t(x) = (\eta_1^t(x), \eta_2^t(x), \eta_3^t(x)), \eta_4^t(x)) = (S(t; x), I(t; x), Q(t; x), R(t; x))$ corresponds to the output of the SIQR model with $x$ representing parameters of rate functions $\{\beta(t; x), \gamma(t; x), \delta(x), \gamma(x)\}$. By our assumption, $d^t|\eta^t(x) \sim \mathcal{N}(\eta^t(x), 1)$, and the Maximum Log-likelihood Estimation (MLE) of $x$ corresponds to [28], [30]:

$$x^* = \text{argmax}_x \sum_{t}^{T} g(\eta^t(x)),$$

$$g(\eta^t(x)) := -\frac{1}{T} \sum_{i=1}^{4} \left(d_i^t - \eta_i^t(x)\right)^2, \quad (3)$$

where $g$ denotes the likelihood function as the negative Square Error (SE) function based on the Gaussian assumption. For notation compactness, we denote $g(\eta^t(x))$ as $f^t(x)$. The goal of model calibration is the maximization of the corresponding negative Mean Square Error (MSE), $\sum_t f^t(x)$.

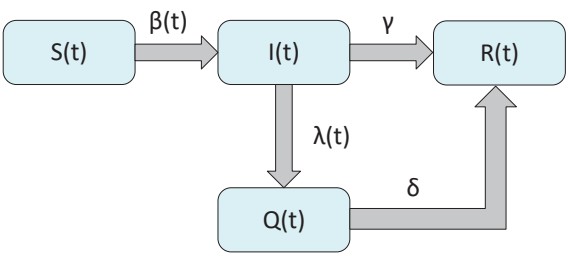

Fig. 1: Diagram depicting an SIQR compartmental epidemiological model. Each arrow indicates that the population rise of the ending compartment is caused by the population decrease of the starting compartment.

Fig. 2: Function network structure of the SIQR model calibration, where the two bold arrows indicate that $x$ and $g$ are the parent node and child node of $y_{1:4}$ respectively.

### III. METHODOLOGY

Instead of following the traditional KOH Bayesian model calibration scheme [20], we resort to a new model calibration framework leveraging Bayesian Optimization (BO) [17]. Under this framework, model calibration is formulated into the optimization of the correspondingly designed utility function, called *acquisition function*. In BO, Gaussian Processes (GPs) have been typically used as cheap surrogates of $f(x)$[1]. In this way, we circumvent the difficulties in optimizing $f(x)$ directly. To be more specific, the standard BO approach treats $f(x)$ as a blackbox function and starts modeling it by a set of GP prior distributions, $\mathcal{N}(\mu(x), \Sigma(x))$. Then, it iteratively chooses the next point at which to evaluate $f$ with the procedure described as follows.

Given $n$ observations of the objective function customized for model calibration, $f(x^1), \ldots, f(x^n)$, we first infer the posterior distribution, $\mathcal{N}(\mu_n(x), \Sigma_n(x))$, which conditions on $\{f(x^l), x^l\}_{l=1}^n$. Then, this posterior distribution is used to compute an acquisition function. Finally, we make a *Bayesian decision* that chooses the next point to be evaluated, $x^{n+1}$, as the point that maximizes this acquisition function. After $N$ iteration, the candidates $x^{best} = \operatorname{argmax}_x \mu_N(x)$ is the calibrated parameter choice.

#### A. Graybox BO

It is clear that given the simulated dynamics $\eta(x)$, evaluation of the model calibration performance metric $g$ is cheap and does not need to be approximated. Instead of directly modeling $f(x)$ as a blackbox objective function as typically done in BO, for SIQR model calibration, we here first use a set of GPs, denoted by $\mathbf{y}(x)$, as the surrogate to $\eta(x)$. With that, we then model $f(x)$ as a graybox objective, by a *Composite Function (CF)* [5]:

$$g(\mathbf{y}(x)) = \frac{1}{T} \sum_i (d_i - y_i(x))^2. \tag{4}$$

---

[1]In the following text, we will consider the maximization of the acquisition function $f^t(x)$ instead of $\sum_t f^t(x)$ for notation compactness, and omit the subscript $t$, where it does not introduce ambiguity.

Spared from the modeling of $g$, the model calibration performance can be expected to be further improved. Furthermore, as shown in Figure 1, the workflow of the SIQR model encompasses a set of input-output structures between functions of each compartment. For example, infectious population 'I' are exclusively transferred from susceptible population 'S', and individuals in the recovery compartment 'R' only come from compartments 'I' and 'Q'. Therefore, one may expect that leveraging these functional structures may lead to improved calibration performance. Integrating these relationships into the BO method gives rise to a *Function Network (FN)*, $\mathcal{G}(\mathcal{V}, \mathcal{E})$, as shown in Figure 2. In the graph, the set of nodes $\mathcal{V} = \{v | y_1, \ldots, y_4, x, g\}$ denotes four GP sets associated with the four compartments 'S', 'I', 'Q' and 'R' respectively, calibration parameters $x$, and performance metric $g$. The graph also contains a set of edges $\mathcal{E} = \{(v \to v') | v, v' \in \mathcal{V}\}$, where $(v \to v')$ means node $v'$ takes the output of nodes $v$ as its input [6]. Suppose we already have made queries from the computer model $\eta$ by simulating trajectories of subpopulations in each compartment at different settings and obtain $\mathcal{D}_n = \{(\hat{\mathbf{y}}^l = \eta(x^l), x^l)\}_{l=1}^n$ at $n$ calibration parameter vectors. We have:

$$P_n(\mathbf{y}(x)) = \prod_{i=1}^4 P_n(y_i(x, Pa_i)), \tag{5}$$
$$y_i \sim \mathcal{N}\left(\mu_n^i(x, Pa_i), \sigma_n^i(x, Pa_i)\right),$$

where $Pa_i = \{y_j : (y_j \to y_i) \in \mathcal{E}\}$ denotes the GPs that are the parent nodes of $y_i$ in graph $\mathcal{G}$, $P_n(y_i(x, Pa_i)) = P(y_i(x, Pa_i) | \hat{\mathbf{y}}^{1:n}(x^{1:n}))$ denotes the posterior distribution of GP $y_i$ with mean $m_n^i$ and variance $\sigma_n^i$. Given $\mathcal{D}_n$, we choose the GP prior $P\left(y_i^{1:n}(x^{1:n}, Pa_i^{1:n})\right)$ to be multivariate normal with a Matérn covariance kernel and a constant mean, whose hyper-parameters are obtained by MLE over $P(\hat{y}_i^{1:n}(x^{1:n}, \widehat{Pa}_i^{1:n}))$. Then we infer the posterior from $P(\hat{y}_i^{1:n}(x^{1:n}, \widehat{Pa}_i^{1:n}), y_i(x, Pa_i))$ [15].

Taking the Knowledge Gradient (KG) [15], [16], a typical formulation of acquisition functions as an example, we define

the graybox acquisition function according to the FN as:

$$\alpha_n(x) = \mathbb{E}_{P_n(\mathbf{y}(x))}\left[u_{n+1}^*(\mathbf{y}, x)\right] - u_n^* \qquad (6)$$

where $u_n^* = \max_x u_n(x)$, $u_n(x) = \mathbb{E}_{P_n(\mathbf{y}(x))}[g(\mathbf{y})]$, $u_{n+1}^*(\mathbf{y}(x)) = \max_{x'} u_{n+1}(x'; \mathbf{y}(x))$, and $u_{n+1}(x'; \mathbf{y}(x)) = \mathbb{E}_{P_{n+1}(\mathbf{y}'(x'); \mathbf{y}(x))}[g(\mathbf{y}')]$. On the right-hand side of the formula above, the second term is the largest expected metric value based on the current GPs, and the first term is the largest expected metric value if the GPs are further conditioned on predicted data $(\mathbf{y}(x), x)$. In comparison, the expectation $\mathbb{E}_{P_n}[\mathbf{y}]$ and $\mathbb{E}_{P_{n+1}}[\mathbf{y}]$ are computed in the original KG.

Due to the maximization operator within the expectation, neither the original nor graybox versions of KG can be computed explicitly. Thus, we resort to Monte Carlo estimation [3] of the expectations and the reparametrization trick [21], so that $\hat{\alpha}$ is an unbiased estimation of $\alpha$ and $\nabla_x \hat{\alpha}$ is feasible. To be more specific, we estimate the acquisition function via:

$$\hat{\alpha}(x) = \frac{1}{K}\sum_{k=1}^{K}\hat{u}_{n+1}^*\left(x, \hat{\mathbf{y}}_{(k)}(x)\right) - \hat{u}_n^*,$$

$$\hat{u}_{n+1}^*\left(x, \hat{\mathbf{y}}_{(k)}(x)\right) = \max_{x'}\frac{1}{L}\sum_{l=1}^{L}g\left(\hat{\mathbf{y}}'_{(l,k)}(x'; x)\right),$$

$$\hat{u}_n^* = \max_{x''}\frac{1}{L}\sum_{l=1}^{L}g\left(\hat{\mathbf{y}}''_{(l)}(x'')\right),$$

$$\hat{y}_{i,(k)}(x) = \mu_n^i\left(x, \widehat{Pa}_{i,(k)}(x)\right) + \sigma_n^i\left(x, \widehat{Pa}_{i,(k)}(x)\right)\hat{\epsilon}_i^{(k)}$$

$$\hat{y}'_{i,(l,k)}(x'; x) = \mu_{n+1}^i\left(x', \widehat{Pa}'_{i,(l,k)}(x'); \hat{\mathbf{y}}_{(k)}(x)\right)$$
$$+ \sigma_{n+1}^i\left(x', \widehat{Pa}'_{i,(l,k)}(x'); \hat{\mathbf{y}}_{(k)}(x)\right)\hat{\epsilon}_i^{(l,k)}$$

$$\hat{y}''_{i,(l)}(x'') = \mu_n^i\left(x'', Pa''_{i,(l)}(x'')\right) + \sigma_n^i\left(x'', Pa''_{i,(l)}(x'')\right)\hat{\epsilon}_i^{(l)},$$
$$(7)$$

where $\hat{\epsilon}_i$ denotes a sample of $\epsilon_i \sim \mathcal{N}(0,1)$, $K = 8$, and $L = 128$. The inference of $P_n$ mainly takes account of the computation cost, which is about $\mathcal{O}(n^3)$ and does not scale up with the dimension of GPs' input (i.e. $x$ or $[x, Pa_i]$) [27]. Thus, the computation complexity of the original and graybox versions of KG are similar. The workflow of our calibration method based on graybox BO is illustrated in Figure 3 with the pseudo-code summarized in Algorithm 1.

### B. Decoupled Decision-Making

In practice, epidemiological data $\mathbf{d}$ from the unknown epidemic process $\zeta$ may not be observed at synchronized time points. In other words, some observations at the corresponding time point may be missing in practice. For example, when $d = \{\varnothing, d_2, d_3, d_4\}$, where $d_1$ is missing in the observed data, we may only be able to compute $g(\mathbf{y}(x)) = \frac{1}{T}\sum_{i \neq 1}(y_i(x) - d_i)^2$. In such a scenario of *decoupled* observations, the function network organization of GPs allows us to infer the function form $y_1(x)$ by leveraging the ground-truth functional dependency represented by $\mathcal{G}$. To be more specific,

$$g \not\perp v, \quad \forall v \in \text{An}_g, \qquad (8)$$

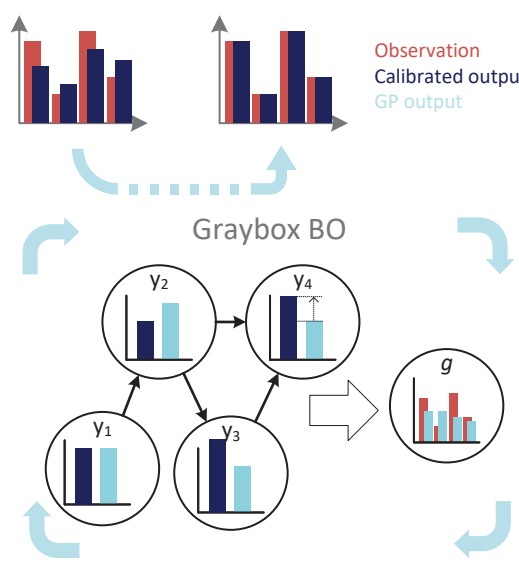

Fig. 3: Schematic illustration of the model calibration workflow based on graybox BO, where we leverage expert knowledge about functional dependency and metric function.

where $v \not\perp v'$ indicates that $v$ and $v'$ are statistically dependent so that $P(v|v') \neq P(v)$ for any two random variables $v$ and $v'$ [22]. This mimics the functional dependency $h(\cdot) \not\perp h'(\cdot)$, which means that a perturbation in function $h(\cdot)$ will affect the solution space of function $h'(\cdot)$; $\text{An}_g = \{v|(v \to \ldots \to g) \in \mathcal{G}\}$, denoting the ancestor nodes of $g$ with a directed path starting at the ancestor node and ending at $g$. The corresponding graph with $d = \{\varnothing, d_2, d_3, d_4\}$ will have the edge $(y_i \to g)$ removed, while $g \not\perp y_1$ still holds.

Moreover, it is reasonable to assume that the approximation of the computer model's output $\{\eta_1(x), \ldots, \eta_4(x)\}$ by the four GPs varies in computational complexity. The reason is that the complexities of the chosen function forms of $\{\eta_1, \ldots, \eta_4\}$ are different. Therefore, some GPs may need more queried data from the computer model than others. Taking into account of the decomposable structure of $P_n(\mathbf{y}(x))$, we extend the graybox acquisition function by *decoupled* decision-making and define the acquisition function as:

$$\alpha_n(x, \mathbf{z}) := \frac{1}{\mathbf{1}^\top \mathbf{z}}\mathbb{E}_{P_n(\mathbf{y}(x))}\left[u_{n+1}^*(\mathbf{z}, \mathbf{y}(x))\right] - u_n^*$$
$$u_{n+1}^*(\mathbf{z}, \mathbf{y}(x)) := \max_{x'}\mathbb{E}_{P_{n+1}(\mathbf{y}'(x'); \mathbf{z}, \mathbf{y}(x))}[g(\mathbf{y}')] \qquad (9)$$

where $\mathbf{z} \in \{0,1\}^4$ and $P_{n+1}(\mathbf{y}'(x'); \mathbf{z}, \mathbf{y}(x))$ is defined as:

$$\prod_{i=1}^{4}P_{n+1}(y_i'(x', Pa_i'); y_i(x, Pa_i))^{z_i}P_n(y_i'(x', Pa_i'))^{1-z_i}.$$
$$(10)$$

New candidates $x^{n+1}$ is found by $\text{argmax}_{x,\mathbf{z}}\alpha_n(x, \mathbf{z})$. When $\mathbf{z} = \mathbf{1}$, $P_{n+1}(\mathbf{y}'(x'); \mathbf{z}, \mathbf{y}(x)) = P_{n+1}(\mathbf{y}'(x'); \mathbf{y}(x))$. The proposed objective can be seen as a generalization of the KG

objective, which measures expected improvement when GPs are conditioned on predicted data $(\mathbf{y}(x), x)$. Here we further accommodate the case when a subset of GPs is conditioned on predicted data. Thus, candidates $x^{n+1}$ with higher expected improvement can be expected by the proposed objective.

Due to the combinatorial nature of $\mathbf{z}$, it is hard to optimize w.r.t. $\mathbf{z}$ directly. The most naive approach is to optimize $\alpha_n$ w.r.t. $x$ under every possible choice of $\mathbf{z}$, and the resulting computational cost is $2^{|\mathbf{z}|}$ times that of the non-decoupled version. For the SIQR model in this paper, $|\mathbf{z}| = 4$ and the increased cost is acceptable. Besides, the non-decoupled version is just a special case of the decoupled version, where $\mathbf{z} = \mathbf{1}$. Therefore, we can accelerate the computation by selecting a subset. In this paper, we choose $\{(\mathbf{z} : z_1 = 1, z_{\neq 1} = 0), \dots, (\mathbf{z} : z_{|\mathbf{z}|} = 1, z_{\neq |\mathbf{z}|} = 0)\} \cup \{\mathbf{z} = \mathbf{1}\}$, resulting $|\mathbf{z}| + 1$ times computation cost of the non-decoupled version.

---

**Algorithm 1** Model Calibration Workflow

---

**Require:** Number of iteration $N$, Computer model $\eta(x)$
  $(X, Y) \leftarrow (X_{init}, \{\eta(x)\}_{x \in X_{init}})$ //$X_{init}$ is a randomly initialized set of calibration parameter vectors.
  **for** $n = \{0, \dots, N-1\}$ **do**
    $\mathcal{D}^n \leftarrow (X, Y)$
    $P_n(y_i(x)) \leftarrow \mathcal{N}(\mu_n^i(x), \sigma_n^i(x))$
    $x^{n+1} \leftarrow \arg\max_x \hat{\alpha}_n(x)$.
    $(X, Y) \leftarrow X \cup \{x^{n+1}\}, Y \cup \{\eta(x^{n+1})\}$
  **end for**
  $x^{best} \leftarrow \arg\max_x \hat{u}_N(x)$

---

## IV. EXPERIMENTAL RESULTS

In this section, we conduct experiments to evaluate and compare the performances of model calibration using various Bayesian Optimization (BO) methods. We consider different setups of observed epidemiological data generated from simulated "ground-truth" models. In our experiments, both the ground-truth model $\zeta$ and the computer model to calibrate $\eta$ are SIQR models. Once a trial of epidemiological data $\mathbf{d}$ is simulated, the ground-truth model is no longer accessible. It is also assumed that we are only accessible to computer models' outputs. The goal is to validate the effectiveness of our graybox BO methods for model calibration in *expensive* scenarios and investigate how our proposed acquisition functions can improve the performance of graybox BO-based model calibration.

To generate data based on the ground-truth SIQR model $\zeta$, we set $\beta^*(t) = 0.9I(t)$, $\lambda^*(t) = 0.1I(t)$, $\delta^* = 0.2$, $\gamma^* = 0.2$. The model to calibrate, $\eta$, takes the same form with undetermined parameters $x$, that is, $\delta(t, x) = x_1 I(t)$, $\lambda(t, x) = x_2 I(t)$, $\delta = x_3$, and $\gamma = x_4$. The initial conditions for both $\zeta$ and $\eta$ are: $S(0) = 0.99 * N$, $I(0) = 0.01 * N$, $R(0) = 0$ and $Q(0) = 0$. We consider two scenarios representing the gap between the ground-truth model and the computer model to calibrate. The first way is to add noise to the ground-truth observations $\mathbf{d}$. The second is to have

different setups of $\eta$ from those of $\zeta$. In our experiments, we set $\lambda^*(t)$ in $\zeta$ as a non-linear function, that is, $\lambda^*(t) = \log([I(t), S(t), R(t)]^\top [0.3, 0.06, 0.12] + 1)$, while $\lambda(t)$ in $\eta$ still takes the linear form mentioned above. The corresponding plots for the 30-day simulations are shown in Figure 4.

We compare calibration performances based on the following BO methods: (i) the standard *Expected Improvement (EI)* in the blackbox BO framework without integrating functional structures, (ii) the standard *Knowledge Gradient (KG)*, (iii) the *Knowledge Gradient with composite function (KG-CF)* , (iv) *Knowledge Gradient with function network (KG-FN)*, and (v) the *Decoupled Knowledge Gradient with composite function (DG-CF)*. All algorithms have been implemented based on the BOTorch package [7] while the latter three are all graybox BO variants. In all our experiments, we have run each calibration method five times from the different random seeds, where each run starts with an initial dataset from the computer model, $\mathcal{D}_0 = \left\{ (\hat{\mathbf{y}}^l = \eta(x^l), x^l) \right\}_{l=1}^{2D+1}$, where $x^l$ is randomly selected in $[0, 1]^D$.

Experiments under all three ground-truth models are performed in complete-observation and incomplete-observation setups. In the latter setup, observations of compartment 'S' are assumed to be unknown, so $d_1$ is not involved with the calibration procedure. This helps to investigate whether the function network organization of GPs allows more reliable model calibration under incomplete (decoupled) observation data $\mathbf{d}$ by leveraging the ground-truth functional dependency.

### A. Complete Observations

The experimental results under the complete-observation setting are shown in Figure 5. In all these experiments, it can be observed that our graybox BO methods, including KG-CF, perform significantly better than the blackbox BO methods, EI and KG. This demonstrates that integrating knowledge of $g$ does help improve the calibration performance. The performance of KG-FN is slightly worse than KG-CF. This phenomenon can be ascribed to the following reasons. With complete observations, all the updating of GPs, $\{y_1, \dots, y_4\}$ that approximate $\{\eta_1, \dots, \eta_4\}$ are well-informed, making the information propagating via introducing statistical dependency less important. Besides, the function network organization of GPs make the acquisition estimator $\hat{\alpha}$ more challenging to calculate reliably. As shown in (7), inputs of each GP include not only $x$ but also random samples of $Pa$, which makes the estimated $\hat{\alpha}$ unstable due to the stochastic nature of inputs, and consequently increases the difficulty of finding the optimal point of $\hat{\alpha}(x)$ in each BO iteration. Last but not least, our proposed DG-CF performs better than KG-CF. This validates that the decoupled acquisition can better inform decision-making by conditioning on the predicted data of a subset of GPs, rather than the whole set of GPs.

### B. Incomplete Observations

The experimental results under the incomplete-observation setting are shown in Figure 6. Again, our graybox BO methods perform significantly better than the blackbox BO methods. It

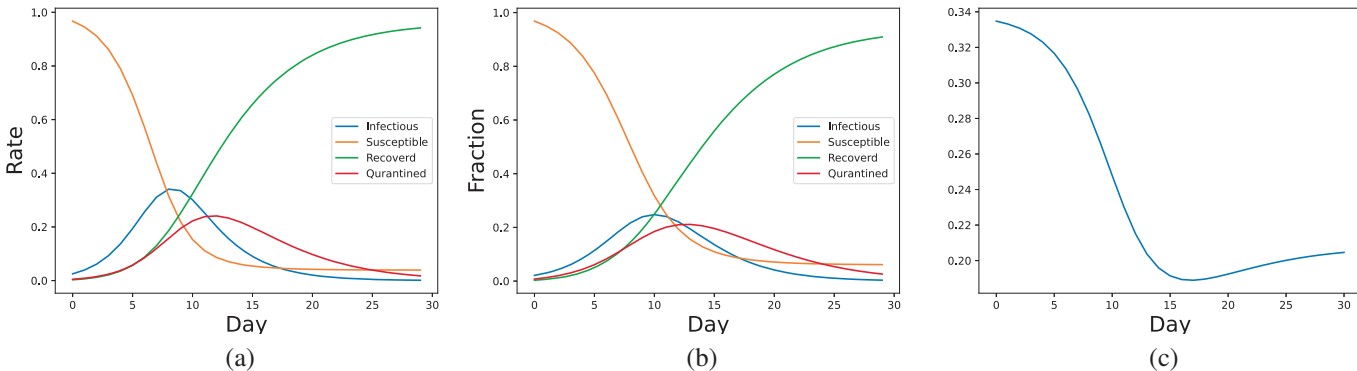

(a)            (b)            (c)

Fig. 4: The simulated ground-truth population fraction trajectories (y-axis) of each compartment for 30 days. (a) Trajectories from an SIQR model with linear derivation functions. (b) Trajectories corresponding to the case when derivation function $\lambda^*(t)$ is non-linear. (c) The trajectory of $\lambda^*(t)$ when it is set to be non-linear.

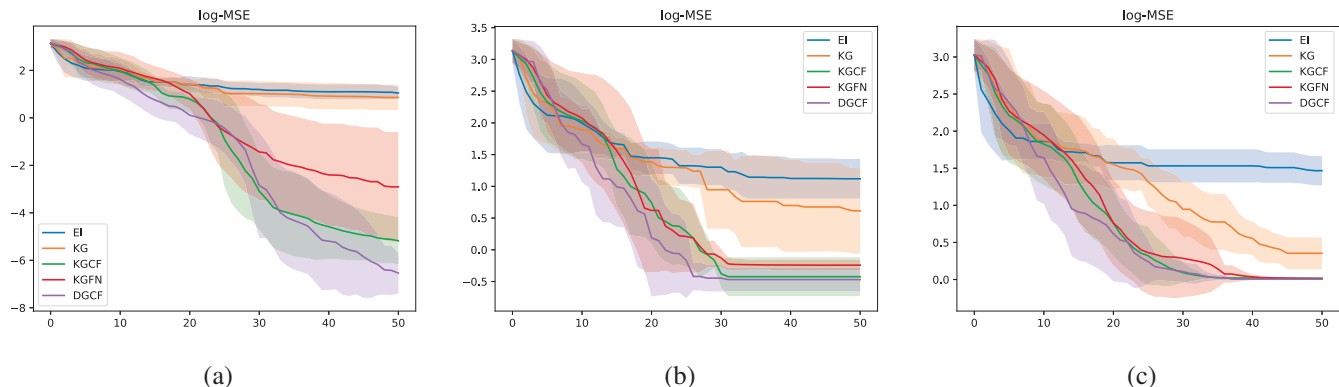

(a)            (b)            (c)

Fig. 5: Calibration performance. The logarithm of the MSE is shown with respect to the number of BO iterations. For the $n_{th}$ iteration, the MSE is computed as $-\sum_t f^t(x)$, where $x = \mathrm{argmax}_x \hat{u}_n(x)$. Solid lines show the mean values and the shaded regions correspond to the standard deviations around the means. (a) Performance for a linear SIQR model. (b) Performance in the presence of noise.(c) Performance for a non-linear SIQR model.

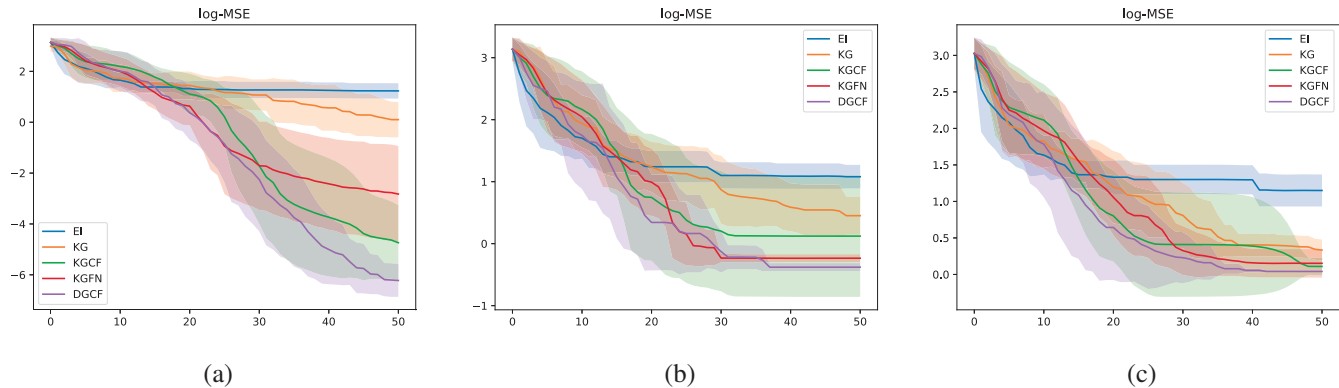

(a)            (b)            (c)

Fig. 6: Calibration performance under incomplete observations. (a) Performance for a linear SIQR model. (b) Performance in the presence of noise. (c) Performance for a non-linear SIQR model. In all cases, we assume that the ground-truth trajectories of the susceptible population (compartment 'S') are missing.

should be noted that KG-FN can outperform KG-CF when the observations are noisy. This shows that integrating functional dependency does help to inform the calibration process, while

the performance gain is not significant due to the accompanied optimization difficulty that we mentioned above. Finally, our proposed DG-CF performs better than KG-CF, validating again

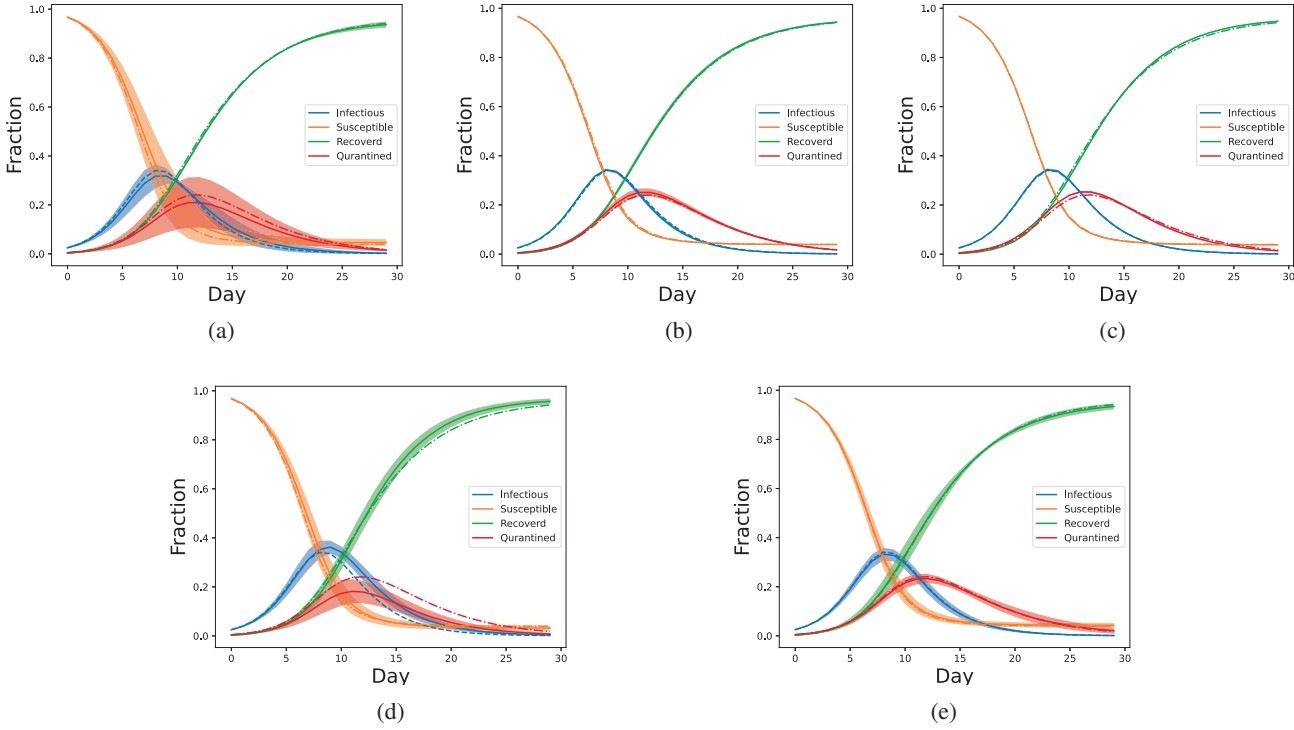

Fig. 7: Simulated population fraction trajectories (y-axis) from the computer model with the calibration parameters $x = \arg\max_x \hat{u}_N(x)$, calibrated by (a) KG-CF, (b) KG-FN, (c) DG-CF, (d) EI, and (e) KG, in the presence of noise. The solid lines represent the mean trajectories simulated from the corresponding calibrated computer models and the shaded regions illustrate the standard deviation intervals around the means. The dotted lines are the ground-truth population fraction trajectories.

the advantages of the decoupled acquisition function.

In Figure 7, We also visualize population fraction trajectories generated from the computer models under the calibrated parameters derived by all methods in the setting with noisy and incomplete observations. It can be observed that all methods can provide non-trivial calibration results, showing that our BO-based calibration methods have a promising potential for epidemiological dynamic model calibration, when the computed model is *expensive*. Among all the BO variants, graybox BO variants KG-FN and DG-CF render the mean trajectories with smaller bias to the ground-truth ones as well as smaller variance, compared to KG-CF, EI, and KG. This again confirms that utilizing the functional dependency structure and decoupled acquisition functions can help better inform decision-making during BO iterations, achieving higher sample efficiency.

## V. CONCLUSION & FUTURE WORK

In this work, we have proposed epidemiological model calibration methods based on the BO framework. To further improve calibration performance, we have formulated the calibration problem as a customized graybox BO task, where expert knowledge about functional dependency and calibration performance metric function is integrated into the new acquisition function. Furthermore, we have proposed a decoupled acquisition function, which further exploits the

decomposable nature of the functional structure. We performed experiments under three types of ground-truth models and two types of observation data to validate the model calibration performance of our proposed BO-based methods. The experimental results have demonstrated the efficacy of our BO-based strategies with different variants for enabling the calibration of *expensive* computer models. Within a small number of BO iterations ($\leq 50$), the proposed graybox BO strategies can achieve good performance, which is measured by logarithm of MSEs, and faster convergence in terms of the number of BO iterations than strategies based on standard BO. While the experimental results have shown that utilizing ground-truth functional dependency can help the calibration process, the resulting formulation of the acquisition function can lead to optimization difficulties in practice, which may impair the overall calibration performance. Thus, future work will focus on further investigation of the function network organization of GPs in BO to achieve better performance. Besides, in the current work, the function network is limited to a single system, where each node corresponds to an element within that system. In a more general case, there can be multiple sub-systems interacting with one another, with each node in the network representing an entire sub-system. Our future investigation will consider extending the proposed calibration strategies to accommodate this more general case. Finally,

agent-based epidemiological models have a much larger number of critical parameters than compartmental models. Along with their significant computational complexity, the underlying functional dependency are also more complex, making their calibration tasks more challenging. Efficient calibration of ABMs for epidemiological modeling may require developing and validating different approximate solutions for BO acquisition functions, which remains an open problem for future research.

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
