# OpenReview forum: "Epidemiological Model Calibration with Bayesian Decision-Making"
_IEEE.org/EMBS/BHI/2024/Conference — IEEE BHI'24_

### Official Review · Reviewer_SHtu · 2024-08-05
**Epidemiological Model Calibration with Bayesian Decision-Making**

**Overall Rating:** 7
**Confidence:** 3

**Other Quality Metrics:**

(a) Clarity of writing Good;
(b) Clinical Significance Good;
(c) Methodological Novelty Great;
(d) Experiments and Results Good

**Questions For The Authors:**

Can you provide more details on the computational resources required for implementing the graybox BO approach? How does it compare to traditional methods in terms of computational cost?

**Strengths:**

The adoption of a graybox BO strategy, which builds upon previous understanding of the functional structure of the model, is a major improvement over conventional blackbox BO techniques.
The paper provides a thorough theoretical framework for the proposed method, ensuring that the approach is well-founded and scientifically robust.
The experimental results emphasise the useful advantages of the suggested approach by demonstrating unequivocally the graybox BO's superiority over the conventional blackbox BO.

**Summary Of The Paper:**

This study introduces a new method of employing Bayesian optimization (BO) to calibrate compartmental epidemiological models, specifically the SIQR (Susceptible-Infectious-Quarantined-Recovered) model. The authors present a "graybox" BO technique that increases calibration efficiency by utilizing the functional structure of the epidemiological models. When this approach is compared with the conventional "blackbox" BO, it shows improved performance. The theoretical framework and experimental results are robust, showcasing the method's efficiency and potential for broader application. However, the paper primarily uses simulated data, and further validation on real-world data is recommended. Despite this limitation, the methodology is innovative. The findings show that the graybox BO technique can be applied to more complicated models, such as agent-based models, and greatly enhances calibration performance.

**Weaknesses:**

The suggested strategy is mainly validated by the study using simulated data. While this is helpful, the method's efficacy would be better demonstrated if it were applied to actual data.
The technique relies on specific statistical assumptions (such as Gaussian noise), which could not always hold true in practical situations and could therefore compromise the precision and stability of the model calibration.
Although the graybox BO method is designed to be efficient, the paper does not provide a detailed analysis of its computational complexity compared to other methods.

---

### Official Review · Reviewer_ra8u · 2024-08-12
**Acceptable after revision.**

**Overall Rating:** 6
**Confidence:** 5

**Other Quality Metrics:**

Good work, but acceptable after revision.

**Questions For The Authors:**

Please see the comments in the "Weaknesses" box.

**Strengths:**

To improve calibration performance,  the calibration problem was formulated as a customized graybox BO task, where expert knowledge about functional dependency and calibration performance metric function and is integrated into the acquisition function.

**Summary Of The Paper:**

This research work has presented and investigated an epidemiological model calibration method based on the bayesian optimization (BO) framework. The results demonstrate the efficacy of the BO strategies with different variants for calibrating expensive computer models. The idea and concept of this work are interesting for presentation in the conference. However, the following comments can be addressed to improve the quality of the work.

**Weaknesses:**

1) the title seems unclear, please add some applications of the proposed work in this part to make it more comprehensive.
2) In the abstract section it is mentioned “we focus on developing efficient calibration methods via Bayesian decision-making for the family of compartmental epidemiological models”, the proposed method can be briefly discussed. To make space, the first two sentences of this section can be removed.
3) Abstract can be supported with some numerical achievements.
4) More keywords can be added.
5) The working principle of the SIQR compartmental epidemiological model and function network structure of the SIQR model calibration shown in Figs. 1 and 2 can be elaborated in depth.
6) How authors have extracted equations 5, 6, and 7? Please discuss.
7) How authors have obtained the calibration performances plotted in Fig.5?
8) Please explain how authors have achieved the experimental results.
9) Please add some numerical results to the conclusion.

---

### Official Review · Reviewer_Vxnz · 2024-08-16
**Accept**

**Overall Rating:** 7
**Confidence:** 4

**Other Quality Metrics:**

(a) Clarity of writing; good
(b) Clinical Significance; good
(c) Methodological Novelty; good
(d) Experiments and Results; good

**Questions For The Authors:**

None

**Strengths:**

Develop model calibration methods via a decoupled decision-making strategy for BO, which further takes advantage of the functional structure's decomposable nature.

**Summary Of The Paper:**

This paper examined model calibration methods based on a “gray box” Bayesian optimization (BO) scheme, which leverages the functional structure of the compartmental epidemiological model to improve calibration performance.

**Weaknesses:**

None

---

### Decision · Program_Chairs · 2024-09-23

Accept